# Bayesian Inverse Physics for Neuro-Symbolic Robot Learning

**Octavio Arriaga**                                    ARRIAGAC@UNI-BREMEN.DE
*Robotics Research Group, University of Bremen*

**Rebecca Adam**                                    REBECCA.ADAM@DFKI.DE
*Robotics Innovation Center, DFKI GmbH*

**Melvin Laux**
*Robotics Research Group, University of Bremen*

**Lisa Gutzeit**
*Robotics Research Group, University of Bremen; Robotics Innovation Center, DFKI GmbH*

**Marco Ragni**
*Technical University of Chemnitz*

**Jan Peters**
*Technical University of Darmstadt; Hessian.AI; Systems AI for Robot Learning, DFKI GmbH*

**Frank Kirchner**
*Robotics Research Group, University of Bremen; Robotics Innovation Center, DFKI GmbH*

**Editors:** Leilani H. Gilpin, Eleonora Giunchiglia, Pascal Hitzler, and Emile van Krieken

## Abstract

Real-world robotic applications, from autonomous exploration to assistive technologies, require adaptive, interpretable, and data-efficient learning paradigms. While deep learning architectures and foundation models have driven significant advances in diverse robotic applications, they remain limited in their ability to operate efficiently and reliably in unknown and dynamic environments. In this position paper, we critically assess these limitations and introduce a conceptual framework for combining data-driven learning with deliberate, structured reasoning. Specifically, we propose leveraging differentiable physics for efficient world modeling, Bayesian inference for uncertainty-aware decision-making, and meta-learning for rapid adaptation to new tasks. By embedding physical symbolic reasoning within neural models, robots could generalize beyond their training data, reason about novel situations, and continuously expand their knowledge. We argue that such hybrid neuro-symbolic architectures are essential for the next generation of autonomous systems, and to this end, we provide a research roadmap to guide and accelerate their development.

## 1. Introduction

One of the most remarkable features of human cognition is its ability to continuously estimate and interact with the world using limited information (Griffiths et al., 2024; Tenenbaum et al., 2011; Pinker, 1984). Humans can recognize an object after seeing it just once and manipulate objects they have never previously encountered (Gopnik, 2012; Lake et al., 2011). Achieving this level of adaptability is critical across the full range of robotic applications. The next generation of industrial and service robots must interact seamlessly with humans and adapt to dynamic surroundings (Lake et al., 2017; Rabinowitz et al., 2018). In healthcare, for example, assistive robots could infer patient needs with minimal supervision, helping to alleviate the

growing demand for medical care in an aging world population (Łukasik et al., 2020; Macis et al., 2022). The need for autonomy with scarce data is even more pronounced in extreme settings. Disaster response robots must navigate unpredictable environments, underwater systems must deal with shifting currents and poor visibility, and space robots face large communication delays and harsh, unpredictable conditions. All of these domains require the ability to reason under uncertainty and adapt autonomously to new circumstances (Nesnas et al., 2021; Obura et al., 2019). Despite all advances in deep learning, current state-of-the-art robot learning lacks these capabilities. Most state-of-the-art algorithms in computer vision and robot learning rely on methods that require massive training datasets with billions of samples or years of simulated experience to solve a set of constrained predefined tasks (Lake et al., 2017). These models struggle to adapt across tasks and environments, making them inefficient and limiting their real-world applicability (Gigerenzer, 2022; Pearl, 2018; Marcus and Davis, 2019; Sünderhauf et al., 2018). Moreover, relying on data-driven black-box models is not only inefficient when known physical approximations exist, but also raises critical safety concerns when robots operate and collaborate in close proximity to humans (Valdenegro-Toro, 2021). These challenges raise two critical questions:

- Why should we ignore the most accurate predictive models that we have of our world?

- Why should we trust the probabilities of a model that we do not understand?

To answer these fundamental questions, this paper argues for an alternative approach that extends current learning and robot perception algorithms by combining and scaling the following ideas:

1. **Uncertainty-aware models** through Bayesian inference, posterior sampling, and variational inference (Ghahramani, 2015; Jaynes, 2003; Parr et al., 2022).

2. **Physics-informed modeling**, combining differentiable physics and graphics-based simulators with neural networks to enhance generalization, efficiency, and physical consistency (Toussaint et al., 2018; Arriaga et al., 2024; Lutter et al., 2020; Heiden et al., 2021).

3. **Program synthesis**, building higher model abstractions with neural program search and transdimensional sampling (Griffiths et al., 2024; Ellis et al., 2020; Cusumano-Towner et al., 2020).

The synthesis of these components provides a direct path to address our previous questions, paving the way for more capable and safe autonomous systems.

## 2. Limitations of Foundation Models for Robotics

In the following section, we present three positions regarding the limitations of foundation models applied to robotics. Those limitations include sample and model complexity, safety, and lack of integrated physical knowledge. However, foundation models have significantly contributed to robotics, including perceptual feature extraction (Oquab et al., 2023), zero-shot instance segmentation (Kirillov et al., 2023), and language models (Dubey et al., 2024). Our conclusions ultimately point to an integrative approach combining foundation models, uncertainty quantification, and integrated physical and symbolic knowledge.

## 2.1. Large Data and Model Complexity

- Performance increases with more data, yet it remains insufficient for robot autonomy.

The transformer models' performance strongly depends on the number of samples and the model size (Kaplan et al., 2020). Experimental studies have shown that this increase in model performance follows a power-law scaling relationship for both data and the number of parameters (Hoffmann et al., 2022; Kaplan et al., 2020). However, both extensive data and large model sizes constitute two of the biggest challenges in using neural networks for robotics. Specifically, there is no internet-scale repository of human-generated robot data, making large-scale learning challenging. Additionally, achieving full autonomy while preserving user privacy necessitates real-time models that run on-device without reliance on external servers. Moreover, highly dynamic and unpredictable environments, such as those found underwater or in space, often lack any pre-existing data, forcing systems to operate under completely unknown conditions.

Current frontier LLMs are estimated to have increased their training data 100-fold since 2020, up to tens of trillions of tokens in 2024 (Brown et al., 2020; Dubey et al., 2024). Moreover, Cottier et al. (2024) estimated computational costs for training LLMs to have increased between 50- and 100-fold since 2020, from hundreds of thousands of USD up to tens of millions in 2024, ultimately resulting in a total estimated cost of more than 100M USD for current frontier models such as GPT4 and Gemini 1.0 Ultra (Cottier et al., 2024). While current frontier models are closed source, and their exact training data and size remain undisclosed, the latest open-source models, such as Llama 3, were trained using 15.6 trillion tokens using a transformer model with 405 billion parameters (Dubey et al., 2024). Two of the latest open-source vision-language-action (VLA) models for robotics, OpenVLA (Kim et al., 2024) and RT-2-X (Collaboration, 2024), consist of 7B and 55B and were trained on 970K and 1M+ demonstration trajectories, respectively. Although trajectories and tokens are not comparable one-to-one, the order of magnitude between both these types of datasets remains considerably large. Furthermore, these robot datasets ignore torque control and operate only in the task space (Kim et al., 2024; Brohan et al., 2022); thus, limiting the applicability to many other robotics tasks that require force feedback.

The success of foundation models relies on web-scale datasets accumulated over decades. Scaling human-collected datasets to that same order of magnitude remains impractical for robotic applications. The limitations of current robotics datasets become particularly acute when considering the goal of enabling robots to perform multiple tasks across diverse environments, while collaborating with other humans and agents. As illustrated in Figure 1, targeting generalization across these multiple axes at once becomes infeasible due to the combinatorial explosion of required data. Current foundation models tend to succeed on only one axis of generalization at the expense of others. For instance, while a segmentation model like SAM2 shows impressive performance across many environments, it is fundamentally limited to a single task (Kirillov et al., 2023; Ravi et al., 2020). Conversely, a model like the Robot Transformer (RT-1) can perform multiple tasks, but its generalization was demonstrated in only three similar kitchen environments and lacked any interaction with humans or other robots (Brohan et al., 2022). This challenge is most evident with real world applications such as self-driving cars. Despite collecting data for years on a scale far beyond typical robotics datasets, self-driving systems have still not achieved Level 5

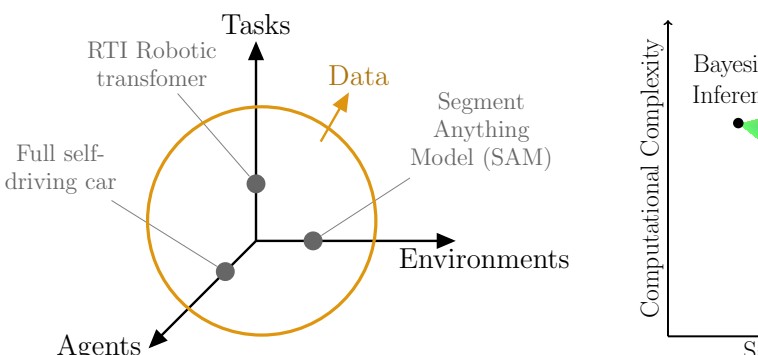 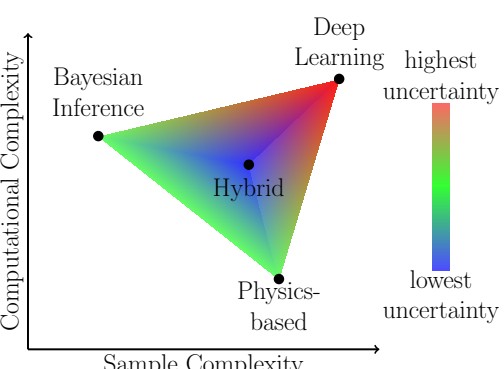

Figure 1: Left: Scaling deep learning models (e.g., SAM, fully self-driving cars, robot transformers) across agents, tasks, and environments dramatically increases data demands, which emphasizes the need for hybrid approaches that integrate physics and symbolic reasoning. Right: Combining Bayesian inference, deep learning, and physics to a hybrid approach optimally minimizes uncertainty and yields the best achievable compromise balancing sample and computational complexity.

autonomy, and their most advanced deployments remain constrained to a small number of geographic locations worldwide.

Recent estimates predict that current LLMs will run out of human-generated data within the next five years (Villalobos et al., 2022). An apparent solution to this problem in robotics is to generate vast quantities of synthetic data from physical simulators. However, in the following sections, we argue that using data to optimize an uninterpretable function approximator is inefficient when a known physical approximation exists and could be explicitly embedded in the model. Constraining the model to be physically plausible not only reduces sample and model complexity but also provides the safety guarantees essential for human-robot collaboration, particularly when integrated within a Bayesian framework for uncertainty quantification.

## 2.2. Trust and Safety

- We cannot trust the probabilities of a model that we do not understand.

Current deep learning (DL) models are known to hallucinate and produce overconfident predictions, often extrapolating incorrect assumptions based solely on correlations in the training dataset (Zhang et al., 2017; Ji et al., 2023). This limitation stems from the fact that DL models typically lack an inherent mechanism to estimate their uncertainty, making them prone to confidently incorrect outputs. Most likely, robot foundation models will inherit these issues in deep learning. This issue is particularly concerning in robot learning, as the robot behavior must generalize to new tasks and environments, i.e., to unseen and out-of-distribution examples. While search and fact-checking can mitigate hallucinations in language models, the consequences in robotics are far more severe. In contrast to errors

in LLMs, where users may correct mistakes afterward, failures in robotic systems occur in real-time and can have irreversible consequences. For example, errors in high-torque systems can lead to immediate fatal consequences, particularly in scenarios where robotic systems operate in direct collaboration with humans.

A significant challenge with deep learning is that it relies on probabilistic outputs from black-box models, which could be wrong in a principled manner by only extrapolating incorrect assumptions found only through correlations in the training dataset. This lack of interpretability raises concerns about safety and the overall trustworthiness of autonomous robotic systems. As these systems become increasingly autonomous through advanced learning, ensuring their safe operation becomes critical. However, autonomy must not come at the cost of human control (Shneiderman, 2020); instead, it should integrate mechanisms that enhance controllability and predictability. This increase in autonomy requires, therefore, coupling with an increase in human control to ensure that robots behave controllably, predictably, and in a comprehensible manner Shneiderman (2020). Evaluating reliability solely based on task success rates is insufficient; instead, trust requires the model's ability to extrapolate causally and measure its uncertainty. Estimating and communicating uncertainty and learning causal world models are the foundations for safe and trustworthy AI.

### 2.3. Physics Unawareness

- Learning algorithms shouldn't ignore the most accurate predictive models of our world.

The most promising large data collection solution for robotics is through simulated data. However, we argue that it is inefficient to use a physics function $p$ to simulate millions of state-action pairs $(s_i, u_i)$ to train a black-box function $f$ to approximate the original function $p$. In many applications of deep RL (DRL) to robotics (Rudin et al., 2022), as well as current robot foundation models (Brohan et al., 2022, 2023; Kim et al., 2024) the known physics are partially ignored and instead learned in combination or parallel with the task itself. With this approach, the learning challenge is made significantly more difficult and valuable resources such as samples and computation are wasted to approximate already known quantities. Especially the notoriously high sample complexity of DRL could greatly benefit from a more principled integration of known physics and lead to better applicability of DRL methods in the realm of robotics (Dulac-Arnold et al., 2021; Ibarz et al., 2021; Kober et al., 2013). A more reasonable alternative is to directly use the physics function $p$ to solve the task, given that physics is the most accurate predictive model of the world, with a short model description, and a strong generalization across domains (Fetter and Walecka, 2003; Landau and Lifshifts, 1960; Feynman et al., 1965). The direct application and approximation of the function $p$ have resulted in optimal control methods and differentiable ray tracers, capable of performing highly athletic behaviors (Kim et al., 2019; Bergonzani et al., 2023; Dai et al., 2014; Javadi et al., 2023; Vyas et al., 2023), and precise scene reconstructions (Nicolet et al., 2021; Loubet et al., 2019). Optimal control models can solve high-level tasks by optimizing at a high frequency the joint torques that control the full rigid body dynamics of the robot, and in some instances elements in the environment (Sleiman et al., 2021). However, there is a detachment between state-of-the-art perception models and action models. Current state-of-the-art computer vision models optimize large-scale neural networks with vast amounts of data, which can be applied to unseen environments and tasks (Bommasani et al., 2021). This

paradigm has provided enormous contributions to the field of computer vision, leading to foundation models in instance segmentation (Kirillov et al., 2023), feature extraction (Oquab et al., 2023), and monocular depth estimation (Yang et al., 2024). However, contrary to the previously described physics-based action models, most computer vision algorithms often ignore the best known physical approximations of the world (Lake et al., 2017). Specifically, most deep learning architectures disregard any information about the interaction between light and matter, even though they are tasked to solve problems that explicitly depend on this physical phenomenon (Spielberg et al., 2023).

## 3. Potentials of Bayesian Inference, Physics and Program Synthesis

### 3.1. Differentiable Physics for Perception and Action

The field of computer graphics has researched the appropriate level of approximation of physics-based rendering functions $p$ that balance computational resources and physical realism (Pharr et al., 2023). Recent advances in computer graphics have led to the development of differentiable physics-based graphics engines (Nimier-David et al., 2019; Laine et al., 2020; Ravi et al., 2020) capable of reconstructing objects from multiple views (Liu et al., 2019; Loubet et al., 2019; Nicolet et al., 2021; Vicini et al., 2022; Arriaga et al., 2024). Furthermore, we can use a differentiable physics function to optimize its inputs using gradient descent to find a desired state. Optimizing the control inputs of a robot system leads to a large set of optimal control algorithms, including trajectory optimization (Betts, 2010; Kelly, 2017) and model predictive control (MPC). Moreover, new developments in differentiable hydroelastic contact models (Elandt et al., 2022; Masterjohn et al., 2022) could enhance the optimization of contact-invariant methods (Mordatch et al., 2012b,a). These methods have shown robots performing complex tasks using high-level descriptions. Furthermore, differentiable physics simulators have allowed building robot algorithms that learn to use tools in order to satisfy their objectives (Toussaint et al., 2018). The broad applications of differentiable physics engines have resulted in a large set of available physics simulators (Todorov et al., 2012; Degrave et al., 2019; Heiden et al., 2021; Geilinger et al., 2020; Freeman et al., 2021). These differentiable physics engines have been combined with neural networks in order to provide more accurate dynamics or close the simulation reality gap (Heiden et al., 2021).

In addition to optimal control, a second promising paradigm for robot control is reinforcement learning (RL) (Sutton and Barto, 2018). Similar to optimal control, RL tackles the problem of sequential decision-making in dynamic systems. A notable difference between RL and optimal control is that optimal control methods assume a known dynamics model, whereas RL aims to learn to make decisions from experienced interactions without making any assumptions about the system dynamics. One promising subfield of RL in particular that could greatly benefit from differentiable physics engines is model-based RL (MBRL) (Moerland et al., 2023). MBRL, either a priori knows or learns a (partial) dynamics model of the environment to speed up learning by allowing the learning algorithm to evaluate long-term effects of the agent's actions based on its approximation of its environment. By enabling the learning RL agent to plan with a model of its environment, the sample-efficiency of RL algorithms can be significantly improved, however, at the cost higher computational requirements. Using computationally efficient and physically sound differentiable dynamics models could further mitigate the negative effects of this trade-off. Moreover, MBRL algo-

rithms with access to a differentiable simulator can compute first-order gradients induced by the policy (first-order MBRL (FO-MBRL) (Freeman et al., 2021; Xu et al., 2022; Georgiev et al., 2024)).

Most of the previous models in both differentiable rendering and differentiable robot simulators do not require additional *training* data or black-box functions to perform dynamic high-level tasks or reconstruct new scenes. Current physical descriptions are accurate and fast enough to be optimized to solve a task under reasonable time constraints. While the lack of data can be advantageous, this implies that the model representation of the system and the environment remains close. While tracking algorithms can compensate for partially unknown dynamics, the environment remains completely predefined. To extend models to deal with dynamic environments and physical changes, we need to consider models that are capable of estimating their uncertainty (Sünderhauf et al., 2018) and expand their model description (Tenenbaum et al., 2011). In the following sections, we describe these two elements in more detail.

### 3.2. Bayesian Inference for Uncertainty-Aware Decision Making

Understanding the physical world requires not only solving inverse problems but also quantifying uncertainty, making Bayesian inference a powerful framework for integrating prior knowledge, refining models with data, and ensuring robust, uncertainty-aware predictions in physics-based applications. A physical simulation $p$, such as an image renderer or a robotic simulator, uses model parameters to describe its dynamics. In the case of image rendering, those parameters could include the positions of the mesh vertices or the location and intensity of light sources. The *forward* problem consists in predicting a measurement given these parameters. Following our computer graphics example, this would render an image given the mesh vertices and the light parameters. The *inverse* problem consists in predicting the parameters given a measurement; thus, predicting the vertices and the light parameters from an image or a set of images. Inverse problem solving may result in solutions where many model parameters describe the same observation (Tarantola, 2005) and need to be constrained by using their physical description or additional information. The Bayesian workflow (Gelman et al., 2020; Martin et al., 2022) allows us to encode uncertainty and knowledge in our system and solve inverse problems through Bayes's theorem (Jaynes, 2003). Computing all possible solutions of an inverse problem can be computationally expensive; thus, instead one can resort to computing only the most likely single solution through maximum a posteriori estimation (MAP) (Murphy, 2022).

An often disregarded application of automatic differentiation is its use in probabilistic programming frameworks. Specifically, the availability of the physical simulations derivatives opens the possibility of using efficient inference algorithms such as Hamiltonian Monte Carlo (HMC), which require the gradient of the target distribution (Betancourt, 2017). Furthermore, we can use approximate and variational inference (VI) to minimize the Kullback-Leibler divergence between the approximate and true distributions (Zhang et al., 2018). Other popular posterior approximations, such as Laplace's approximation, require computing Hessian matrices. Finally, the maximum-entropy principle can be formulated as an optimization problem to build prior distributions using gradient descent (Jaynes, 2003). The widespread adoption and development of automatic differentiation engines in specialized hardware, such

as GPUs, recently bootstrapped the interest in large-scale Bayesian inference frameworks. Some of these new probabilistic programming frameworks include: tensorflow-probability (Dillon et al., 2017), gen (Cusumano-Towner et al., 2019), numpyro (Phan et al., 2019), and blackjax (Cabezas et al., 2024). In general, derivatives are not only useful for computing point estimates of large black-box function approximators, but also to compute the uncertainty of physically grounded models. Uncertainty quantification as well as the direct application of a differentiable physical model can help increase trust and safety in our models, reducing the sample and model complexity of learning algorithms, and building models that can learn continuously Sünderhauf et al. (2018). Bayesian inference, though computationally complex, requires fewer samples and parameters than deep learning by leveraging priors and known models. Integrating it with physics and deep learning offers a balanced tradeoff for uncertainty-aware and complex learning as shown in Fig. 1 on the right.

### 3.3. Integrating Deep Learning and Program Synthesis to Model Worlds

Most probabilistic programming frameworks focus on static probabilistic graphical models. However, machines that learn and think like humans must be able to learn and expand their model of the world (Lake et al., 2017; Tenenbaum et al., 2011). To facilitate incremental continuous robot learning, continual deep learning strategies tackle catastrophic forgetting, by either replaying trained tasks and classes (Rolnick et al., 2019), or by partially incrementally adapting the last output network layers (Rusu et al., 2016; Fernando et al., 2017). However, incorporating explicit knowledge into neural networks and evolving it remain open challenges for the neuro-symbolic AI community. A viable option for encoding explicit knowledge in combination with neural networks is through the use of programs (Rule et al., 2020) and program synthesis (Ellis et al., 2020). State-of-the-art approaches in program synthesis automate the generation of functional programs by leveraging domain-specific languages (DSLs), a program library, and a search strategy (Ellis et al., 2020; Bowers et al., 2023; Grayeli et al., 2024). A DSL provides structured representations of programs, allowing program synthesis systems to explore hierarchical program trees efficiently. However, brute-force searching through these trees is computationally infeasible. Deep learning can manage the tree search in program synthesis to find new programs by scoring, prioritizing, and pruning paths to improve efficiency (Balog et al., 2016; Silver et al., 2016).

Additionally, probabilistic programming can balance program complexity and accuracy by learning optimal model transitions using Reversible Jump Markov Chain Monte Carlo (RJMCMC) (Green, 1995). RJMCMC enables adaptive model selection by jumping between different parameter dimensionalities, naturally favoring simpler, more efficient models. By dynamically exploring both complex and simple solutions, the system can flexibly synthesize programs that meet the required functional specifications while minimizing unnecessary complexity. Initially, computing transdimensional jumps in RJMCMC required manually implementing the proposal kernels and their respective measure-preserving correction factors. Finally, the combination of program synthesis and (deep) RL provides several promising research avenues, e.g., when applying RL for finetuning program synthesis (Qian et al., 2022; Bunel et al., 2018), guiding RL with program synthesis (Yang et al., 2021), or improving policies with program synthesis (Bhupatiraju et al., 2018).

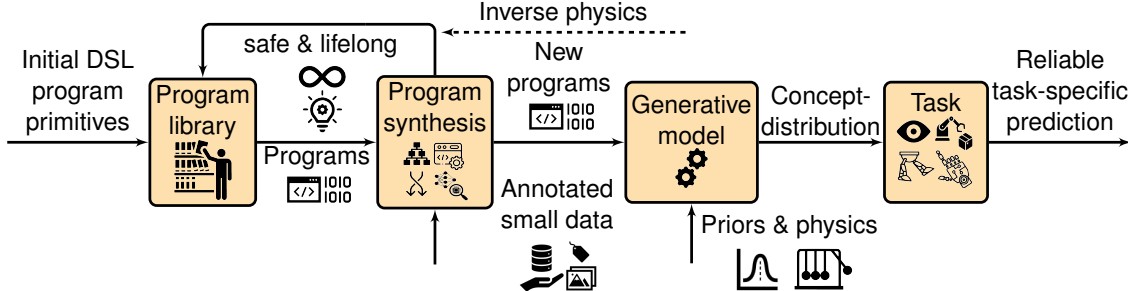

Figure 2: Our neuro-symbolic robot learning framework uses Bayesian inverse physics and program synthesis to solve diverse tasks safely, and physically consistently. Theoretically, this concept continuously evolves knowledge without forgetting.

## 4. Roadmap for Future Research

Our proposed neuro-symbolic roadmap for robotics prioritizes physically-grounded and uncertainty-aware reasoning for reliable real-world interaction. In this framework, deep learning and foundation models are not the primary tool but rather a complement to physics-based models, and are instead used to learn residual components, find data-driven priors, or initialize optimization problems. This integration is fundamental to lifelong learning systems, where methods like program synthesis can continuously expand the initial robot's knowledge. Therefore, our short-term roadmap synthesizes physics, Bayesian inference, and program synthesis to build a framework for adaptable intelligent systems (Fig. 2).

- **Domain-specific languages (DSL) and primitives:** Program synthesis techniques require an initial set of reusable and extensible program primitives. To this end, the first step will include defining a versatile DSL and primitives useful for diverse robotic tasks employing both Bayesian inference and differentiable physical simulators. This new language can reuse or draw inspiration from established frameworks, including probabilistic programming systems like Pyro (Bingham et al., 2019), and automatic differentiation libararies like JAX (Frostig et al., 2018).

- **Differentiable physics:** Integrating differentiable physical simulations of perception and action within machine learning models can reduce the number of parameters and training examples used in neural networks. Moreover, physical consistency can limit the search space of program synthesizers by pruning infeasible solutions. Within this context, we need to integrate multiple physical simulators, including rigid body simulators (Todorov et al., 2012), hydroelastic contact models (Elandt et al., 2022), and differentiable renderers (Nimier-David et al., 2019).

- **Bayesian inference:** Robots need to be capable of reasoning under uncertainty. Specifically, they must estimate their posterior distributions. The second step should involve implementing probabilistic models within the DSL, utilizing probabilistic programming frameworks, and applying variational inference to handle computational complexity (Cabezas et al., 2024; Blei et al., 2017).

- **Deep learning integration:** Naive program synthesis involves searching over a large discrete space. Lacking a differentiable continuous search space, one cannot directly utilize gradient-based optimization. Neural search in combination with current LLMs provides a path forward to alleviate these issues (Balog et al., 2016; Shi et al., 2023; Ellis et al., 2020; Guo et al., 2024).

- **Program synthesis for continual library learning:** Program synthesis presents a viable option to represent multiple domains of knowledge (Rule et al., 2020). By iteratively refining and synthesizing symbolic program libraries and updating knowledge, program synthesis retains old knowledge and is inherently adaptive (Bowers et al., 2023; Ellis et al., 2020; Grayeli et al., 2024). Moreover, recent developments in Reversible Jump Markov Chain Monte Carlo (RJMCMC) (Neklyudov et al., 2020; Cusumano-Towner et al., 2020) open the possibility for Bayesian program synthesis (Saad et al., 2019; Matheos et al., 2020).

Bayesian inverse physics demands extensive simulations and high-dimensional sampling, making inference computationally expensive and challenging for real-time robotic decision-making. Efficient approximations, parallelization, or surrogate models will be essential for practical deployment in dynamic environments and require further research. Building Bayesian models requires careful prior and likelihood selection, limiting automation in model construction. However, future work could explore automating prior selection while ensuring physical consistency. Targeting long-term real-world deployability for the hybrid approach requires more research on novel benchmarking methods and real-world testing. Finally, the seamless integration of symbolic program synthesis, neural networks and differentiable physics engines constitutes a huge challenge and requires ensuring compatibility between discrete symbolic representations and continuous robotic actions.

## 5. Conclusion

Recent advancements in neural networks have attracted significant private investment, fueling the development of generic function approximators. The available vast private data and computational resources risk fostering a research culture focused on ever-larger datasets and black-box models, consuming energy on the scale of nuclear power plants. However, the human brain operates on approximately 20 watts. A more principled approach would be first to encode the physical approximations we already know into our models, serving as a foundation for model-building algorithms inspired by program synthesis and probabilistic programming, enabling more efficient and interpretable learning. While current foundation models are limited in building general-purpose robotics, their specialized skills are valuable for well-defined tasks in closed environments. We propose exploring alternatives that improve generalization, transparency, and safety while reducing data and computational demands.

## 6. Acknowledgments

This work was funded with the Robotics Institute Germany (RIG) (Grant 16ME1010) and the ActGPT (Grant 01IW25002) projects, through the German Federal Ministry of Education and Research (BMBF).

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
