# OpenReview forum: "Bayesian Inverse Physics for Neuro-Symbolic Robot Learning"
_nesyconf.org/NeSy/2025/Conference — NeSy 2025 Poster_

### Official Review · Reviewer_jA1m · 2025-04-04
**Good position paper**

**Rating:** 7
**Confidence:** 4

**Review:**

This position paper provides an in-depth analysis of the state-of-the-art AI and machine learning systems most commonly used in robotics, highlighting three key challenges that are still not fully addressed by current approaches: (i) reducing the data-hungriness of learning systems, given that large datasets are typically scarce in robotic applications; (ii) ensuring that the learned systems are safe and trustworthy; and (iii) exploiting known physical relations governing actions and perceptions into the learning process.

To tackle these challenges, the authors propose a compelling combination of three core components: (i) the integration of differentiable dynamic models; (ii) the use of Bayesian inference to leverage priors and known models, as well as to quantify uncertainty in decision-making; and (iii) program synthesis for evolving explicit knowledge of the world.

The paper is well-written and well-structured, making it a pleasure to read. It offers valuable insights from the perspective of AI applied to robotics. The resulting analysis is both thorough and convincing. For these reasons, I recommend the paper for acceptance.

**Anonymity:**

Remain anonymous

---

### Official Review · Reviewer_VzHV · 2025-04-08
**Very interesting research vision paper**

**Rating:** 10
**Confidence:** 5

**Review:**

The paper outlines several important research directions in the area of Robotics where Neuro-Symbolic approaches will play a significant role. More specifically, it provides a crystal-clear analysis of the limitations of current pre-trained large ML models when applied to robotics, offering a set of convincing motivations and extensive references to the literature.

In the second part, the paper puts forward a set of methodologies and research areas that could help overcome the aforementioned limitations. These areas include Bayesian Inference, Probabilistic Programming, Differentiable Physics, and World Symbolic Modeling.

In the final part, the paper draws a research roadmap that identifies the most important contributions the NeSy community can provide to develop robotic platforms capable of learning and adapting to unseen environments, and of accomplishing tasks they have never encountered before.

I believe this paper is of extremely high quality and provides a set of important stimuli for the NeSy community, going beyond short-term research problems. It clearly describes a vision and is strongly supported by the literature.

I believe that this paper not only deserves to be accepted to the conference, but should also be considered as the basis for a discussion session on the potential contributions of NeSy to Robotics.

**Anonymity:**

Disclose identity

---

### Official Review · Reviewer_M7hD · 2025-04-10
**Interesting paper but too high level**

**Rating:** 6
**Confidence:** 4

**Review:**

In their position paper, the authors propose an approach to neurosymbolic robot learning that integrates physics, Bayesian inference, and program synthesis as an alternative to foundational LLMs. This is certainly an interesting idea to discuss at NeSy. However, the paper itself is very high-level, and it does not give any details on how the proposed architecture can be implemented, let alone a first prototype implementation of the approach or parts of it.

**Anonymity:**

Remain anonymous